# Cognitive Sparing in Proton versus Photon Radiotherapy for Pediatric Brain Tumor Is Associated with White Matter Integrity: An Exploratory Study

**DOI:** 10.3390/cancers15061844

**Published:** 2023-03-19

**Authors:** Lisa E. Mash, Lisa S. Kahalley, Kimberly P. Raghubar, Naomi J. Goodrich-Hunsaker, Tracy J. Abildskov, Luz A. De Leon, Marianne MacLeod, Heather Stancel, Kelley Parsons, Brian Biekman, Nilesh K. Desai, David R. Grosshans, Arnold C. Paulino, Zili D. Chu, William E. Whitehead, Mehmet Fatih Okcu, Murali Chintagumpala, Elisabeth A. Wilde

**Affiliations:** 1Department of Pediatrics, Division of Psychology, Baylor College of Medicine, Houston, TX 77030, USA; 2Psychology Service, Texas Children’s Hospital, Houston, TX 77030, USA; 3Texas Children’s Cancer and Hematology Centers, Texas Children’s Hospital, Houston, TX 77030, USA; 4Department of Neurology, University of Utah School of Medicine, Salt Lake City, UT 84132, USA; 5Department of Psychology, University of Houston, Houston, TX 77204, USA; 6Department of Radiology, Division of Neuroradiology, Texas Children’s Hospital, Houston, TX 77030, USA; 7Department of Radiology, Baylor College of Medicine, Houston, TX 77030, USA; 8Division of Radiation Oncology, The University of Texas MD Anderson Cancer Center, Houston, TX 77030, USA; 9Department of Neurosurgery, Baylor College of Medicine, Houston, TX 77030, USA; 10Department of Pediatrics, Division of Hematology Oncology, Baylor College of Medicine, Houston, TX 77030, USA; 11Department of Pediatrics, Division of Physical Medicine and Rehabilitation, Baylor College of Medicine, Houston, TX 77030, USA

**Keywords:** pediatric brain tumor, proton radiotherapy, white matter, diffusion tensor imaging, late effects

## Abstract

**Simple Summary:**

Research has shown that children who undergo radiotherapy for brain tumors are at risk for long-term changes in both their thinking and brain structure. Compared to photon radiotherapy (i.e., X-rays), proton radiotherapy may cause less damage to healthy brain tissue and result in fewer cognitive problems. This study compared cognitive functioning and white matter damage in survivors of pediatric brain tumors who were treated with proton or photon therapy. The results showed that patients who received photon therapy had more cognitive problems and showed more white matter change than those who received proton therapy. Patients who underwent proton therapy, on the other hand, were similar to healthy individuals with no history of brain tumors. This study suggests that proton therapy may protect healthy brain tissue, leading to better long-term cognitive outcomes.

**Abstract:**

Radiotherapy for pediatric brain tumors is associated with reduced white matter structural integrity and neurocognitive decline. Superior cognitive outcomes have been reported following proton radiotherapy (PRT) compared to photon radiotherapy (XRT), presumably due to improved sparing of normal brain tissue. This exploratory study examined the relationship between white matter change and late cognitive effects in pediatric brain tumor survivors treated with XRT versus PRT. Pediatric brain tumor survivors treated with XRT (*n* = 10) or PRT (*n* = 12) underwent neuropsychological testing and diffusion weighted imaging >7 years post-radiotherapy. A healthy comparison group (*n* = 23) was also recruited. Participants completed age-appropriate measures of intellectual functioning, visual-motor integration, and motor coordination. Tractography was conducted using automated fiber quantification (AFQ). Fractional anisotropy (FA), axial diffusivity (AD), and radial diffusivity (RD) were extracted from 12 tracts of interest. Overall, both white matter integrity (FA) and neuropsychological performance were lower in XRT patients while PRT patients were similar to healthy control participants with respect to both FA and cognitive functioning. These findings support improved long-term outcomes in PRT versus XRT. This exploratory study is the first to directly support for white matter integrity as a mechanism of cognitive sparing in PRT.

## 1. Introduction

Pediatric brain tumors are the leading cause of cancer-related death for children in the United States [1]. Central nervous system (CNS)-directed radiotherapy is often a life-saving treatment, but it has been associated with significant long-term morbidity, including progressive cognitive decline [2,3,4]. As survivorship has improved over recent decades [5], accumulating evidence suggests that cognitive changes may, in turn, impact later psychosocial outcomes in adulthood, such as employment, social attainment, and independent living skills [6,7,8,9]. Elucidating the underlying mechanisms of neurocognitive decline may aid in the selection of treatments that minimize cognitive sequelae and improve functional outcomes.

Conventional radiotherapy places pediatric brain tumor survivors at risk for intellectual decline [2,10], with an estimated 1-point loss in full-scale IQ per year following treatment [11]. This is thought to reflect a plateau in new skill development and has been strongly associated with deficits in underlying cognitive proficiency skills, such as attention, psychomotor speed, and working memory [12,13,14,15,16,17]. White matter maturation supports the development of these abilities [18,19,20], which depend on coordinated activity across distributed brain networks. Weaknesses in cognitive proficiency have been attributed to white matter compromise in neurodevelopmental conditions, such as attention-deficit/hyperactivity disorder [21] and preterm birth [22], as well as acquired conditions, such as traumatic brain injury [23,24] and multiple sclerosis [25]. Thus, reduced white matter integrity is widely accepted as a major mechanism of neurocognitive alteration following insult, including both the primary cause of injury (e.g., displacement or tissue compromise from tumors) and secondary forms of injury from the life-saving treatments to address them, such as CNS-directed radiation.

Progressive white matter deterioration has been observed following CNS irradiation in both human and animal models [26,27,28]. Evidence suggests that early neuroinflammation and microvascular damage cause apoptosis of oligodendrocytes, leading to cascading demyelination over time (for review, see [29]). This process may result in focal white matter lesions, atrophy, and diffuse microstructural change. Microstructural damage may not be apparent on conventional magnetic resonance imaging (MRI) sequences; however, diffusion tensor imaging (DTI) may capture these subtle changes as early as one month following radiotherapy [30].

In humans, myelination continues throughout adolescence and early adulthood; therefore, CNS-directed radiotherapy may be especially detrimental to the immature brain still under development. Indeed, reduced white matter volume and integrity have been demonstrated in children who have undergone radiation for brain tumors [27,28,31]. While these changes are often widespread, some tracts in the developing brain may be especially susceptible to radiation-induced microstructural damage, such as the corpus callosum [32,33,34,35]. Moreover, treatment-related white matter compromise has been associated with deficits in intellectual functioning [10,33,36,37,38,39], attention [12,39,40,41], working memory [42,43], processing speed [10,13,39,44,45], and motor speed [44].

Compared to conventional photon radiotherapy (XRT), proton radiotherapy (PRT) delivers less radiation to healthy brain tissue by reducing the entry dose and eliminating the exit dose [46,47]. Research suggests that PRT is similarly effective but offers improved long-term cognitive outcomes compared to XRT [48,49,50]. Survivors who receive focal (versus craniospinal) PRT demonstrate the clearest advantage, with neurocognitive profiles that are broadly within normative expectations [49,51,52] and similar to patients treated with surgery only [53]. Moreover, unlike those receiving XRT and/or craniospinal radiation, cognitive performance does not appear to decline over time following focal PRT [11,54]. In a longitudinal study of adults treated for glioblastoma, Dünger et al. [55] found that patients who received XRT showed significant reductions in whole-brain mean diffusivity over time on DTI, whereas mean diffusivity was stable over time following PRT. This provides evidence that PRT may cause less long-term white matter compromise than XRT, which may underlie cognitive sparing. To our knowledge, this relationship has not yet been demonstrated in pediatric brain tumor survivors; treatment in this developmentally sensitive period may result in unique neurodevelopmental trajectories. Moreover, white matter damage has not been directly related to differences in neurocognitive outcomes following XRT vs. PRT.

This exploratory study assessed white matter integrity and neuropsychological performance in pediatric brain tumor survivors treated with either XRT or PRT and a sample of healthy controls. Survivors who received XRT were expected to show the greatest reductions in microstructural integrity and to have the lowest neuropsychological test scores. On the other hand, PRT patients were expected to show only mild, if any, declines in white matter integrity and neuropsychological performance compared to healthy controls. Finally, a direct relationship between white matter integrity and neuropsychological performance was anticipated, especially with respect to processing speed and motor coordination.

## 2. Materials and Methods

### 2.1. Participants

Participants were enrolled in this study examining neuroimaging and cognitive outcomes in long-term survivors of pediatric brain tumors. The minimum age of participants at enrollment was 6 years. All participants were fluent in English. Pediatric brain tumor survivors were treated with a single course of PRT or XRT for a primary brain tumor and had no evidence of active disease or recurrence at enrollment. Exclusionary diagnoses included brain stem glioma, high grade glioma, and atypical teratoid/rhabdoid tumors. Due to an institutional change in the standard of care in 2007, all XRT patients were treated between 2000 and 2007 while PRT patients were treated between 2007 and 2013. Eligible participants were identified via medical record review. They were approached by a study coordinator for enrollment between 2018–2019. Of those approached, 78.6% of eligible participants agreed to participate. Participation did not significantly differ with respect to radiation type, sex, race, or tumor histology (all *p* > 0.05). Participation was higher for families identifying as Hispanic (100%) compared to non-Hispanic families (70%; *p* = 0.022). “Other” tumor types not shown in Table 1 included craniopharyngioma, pilomyxoid astrocytoma, desmoplastic ganglioglioma, choroid plexus carcinoma, atypical choroid plexus tumor, and high-grade neoplasm with small blue cell features. Details about tumor type and location for each patient are provided in Appendix A. 

Of those patients who consented to participate (13 XRT, 24 PRT), MRI data was missing or of insufficient quality for 10 individuals (8 PRT, 2 XRT). Two PRT patients did not return for neuropsychological testing. One PRT patient was excluded for a previous course of intensity-modulated radiation therapy. One XRT patient was excluded due to incomplete medical history, having been diagnosed internationally. One PRT patient was excluded due to profound cognitive impairment, which precluded completion of testing. In the right-handed healthy comparison group, 6 participants had no usable imaging data. Altogether, this study reports on the outcomes of 22 patients (XRT *n* = 10, PRT *n* = 12) and 23 healthy controls (CTL), whose characteristics are reported in detail in Table 1. Informed consent was obtained from adult patients or caregivers, and assent was obtained from patients under 18 years of age. This study was approved by the Institutional Review Board at Baylor College of Medicine.

### 2.2. Neuropsychological Measures

All participants completed a battery of standardized neuropsychological tests. Measures were administered in a standardized fashion by trained research assistants under the supervision of a neuropsychologist. Appropriate versions of each test were selected based on participant age. Standard scores (mean of 100, standard deviation of 15) were computed using age-based normative data for all measures.

Intellectual functioning was evaluated using the Wechsler Intelligence Scale for Children (WISC-V, WISC-IV; [56,57]) or the Wechsler Adult Intelligence Scale (WAIS-IV; [58]). Domains assessed included Full-Scale IQ (FSIQ), Verbal Comprehension Index (VCI), Perceptual Reasoning Index (PRI), Working Memory Index (WMI), and Processing Speed Index (PSI). Because the WISC-V does not generate a PRI score, the publisher (NCS Pearson) provided norms to calculate PRI scores to facilitate comparison across the WISC-IV, WISC-V, and WAIS-IV. Reliabilities for the WISC-V PRI ranged from 0.93 to 0.95 for ages 6–16 [59]. Participants also completed the Visual–Motor Integration (VMI) and Motor Coordination (MC) subtests of the Beery–Buktenica Developmental Test of Visual Motor Integration, Sixth Edition [60] The VMI subtest requires participants to copy increasingly complex figures and is scored based on accuracy. The MC subtest measures graphomotor precision by requiring participants to connect dots within sets of lines to form a figure; points are deducted when the examinee marks outside of the lines.

### 2.3. Neuroimaging Procedures

Participants completed at least one 3 Tesla MRI scan (Siemens MAGNETOM Skyra, Erlangen, Germany). Five participants in the PRT group were sedated. A pair of diffusion-weighted images, each with 1 b0 volume and 64 diffusion-weighted directions (b = 1000  s/mm^2^, TR = 4600 ms, TE = 77 ms, in-plane resolution = 0.965 × 0.965 mm, and slice thickness = 2 mm), were acquired, one with an anterior-to-posterior phase encoding direction and a second with a posterior-to-anterior phase encoding direction.

Analyses were completed on a high-performance computing system. Diffusion-weighted DICOM data were converted into NIfTI format using the dcm2niix tool (v1.0.20211006, https://github.com/rordenlab/dcm2niix, accessed on 1 May 2022). Diffusion data were then corrected for susceptibility and eddy current-induced image distortions and co-registered using the TOPUP [61] and EDDY [62,63] functions (i.e., the GPU version “eddy_cuda8.0”) from FSL (version v6.0.5). TOPUP was performed using the default parameters on the opposing phase encoding (i.e., AP and PA) b0 images. EDDY was performed with the replacement of outlier slices [62] and slice-to-volume motion correction [64] and default parameters. The brain mask used for EDDY was obtained by binarizing the sum of the brain masks created from each b = 0 image volume using BET2 function [65] with the fractional intensity threshold set to 0.2.

Deterministic tractography was performed using the open-source Automated Fiber Quantification (AFQ) software package v1.2 (https://github.com/yeatmanlab/AFQ, accessed on 1 May 2022) with default parameters [66] running on MATLAB (R2021b; MathWorks Inc., Natick, MA, USA). Twelve major tracts of interest were selected due to their broad associations with intellectual, cognitive, and motor functions. Tracts included the left and right cingulum bundles (CB), corpus callosum major and minor forceps (CCMa, CCMi), left and right inferior frontal-occipital fasciculi (IFOF), left and right superior longitudinal fasciculi (SLF), left and right inferior longitudinal fasciculi (ILF), and left and right uncinate fasciculi (UNC). The AFQ method tracks fibers in each individual’s native space, resulting in subject-specific tracts. All tracts are shown in a representative control participant in Appendix A. Average fractional anisotropy (FA), axial diffusivity (AD), and radial diffusivity (RD) were estimated for each fiber tract [66]. FA is a broad measure of white matter structural integrity that accounts for the ratio of diffusion along the primary axis (AD) to diffusion perpendicular to the primary axis (RD). Generally, higher AD and lower RD are associated with higher FA. FA ranges from 0 to 1, with higher values indicating proportionally higher diffusion in the primary direction of the axon, which is a marker of structural integrity.

### 2.4. Statistical Analyses

Demographic characteristics and treatment-related variables were compared between XRT, PRT, and CTL groups using one-way analysis of variance (ANOVA) or chi-squared tests of independence. Covariates selected for subsequent analyses included age at evaluation and handedness, which differed between groups. While all XRT and PRT patients were considered to be in “late survivorship” (i.e., minimum 7 years from treatment), time since radiation was longer for all XRT patients than PRT patients due to the institutional change in standard of care described above (i.e., XRT only before 2007, PRT only since 2007). This was not included as a covariate due to redundancy with treatment group; however, supplemental analyses were conducted examining within-group associations between time since radiation and relevant outcome measures to determine whether this may account for observed group effects.

Linear mixed models were used to explore effects of group on white matter integrity (i.e., FA, RD, AD) across all 12 tracts combined. Fixed effects included group and covariates (age at evaluation, handedness). Tracts were nested within subject, which was included as a random effect. Planned contrasts examined differences between XRT and PRT groups, as well as differences between PRT and CTL groups. Tract-level integrity measures and neuropsychological performance were compared between groups using analysis of covariance (ANCOVA), including the same covariates and group contrasts. Pearson correlation was used to assess relationships between neuropsychological performance and FA.

## 3. Results

### 3.1. Demographic and Clinical Characteristics

Demographic characteristics and treatment-related variables are presented in Table 1. XRT, PRT, and CTL groups did not significantly differ with respect to sex, race, ethnicity, maternal education, family income, or household size. Groups differed with respect to handedness (*Χ*^2^(2) = 8.84, *p* = 0.012; 3 left-handed participants in PRT group only), as well as age at evaluation (*F*(2,42) = 4.91, *p* = 0.012; XRT mean = 21.7, PRT mean = 16.9, CTL mean = 15.5). Moreover, XRT and PRT groups did not differ with respect to age at diagnosis, tumor location or type, total radiation dose, number of craniotomies, Karnofsky–Lansky score, or proportion of patients receiving craniospinal irradiation, shunting, or chemotherapy. As described above, time since radiotherapy was consistently longer in the XRT group (mean = 14.7 years) than the PRT group (mean = 8.9 years) as expected.

### 3.2. Group Differences in White Matter Integrity

Tractography was successful for all 12 white matter tracts for 62% of participants (20% of XRT, 67% of PRT, 78% of CTL). No individual participant was missing more than 2 of 12 tracts, and no individual tract was missing data for more than 4 participants. Linear mixed models revealed a significant effect of group on overall FA when covarying for age and handedness, such that lower FA across all tracts was found in XRT versus PRT patients (*β* = −0.027, *t*(514) = −2.58, *p* = 0.010). On the other hand, overall FA did not significantly differ between PRT and CTL groups. Consistent with the FA findings, overall RD was significantly higher in XRT than PRT patients (*β* = 0.035, *t*(514) = 4.21, *p* < 0.001) but did not differ between PRT and CTL groups. No group differences were found in overall AD. Results of linear mixed models are summarized in Table 2.

To examine potential effects of craniospinal irradiation on these findings, supplemental analyses were performed, including only patients who underwent focal radiotherapy (XRT *n* = 5, PRT *n* = 8). Linear mixed models in this subsample of participants yielded the same pattern of results, with the XRT group showing significantly lower overall FA and higher overall RD compared to the PRT group. Again, no significant differences between PRT and CTL groups were found (Appendix A).

Tract-level group differences in FA, AD, and RD are shown in Figure 1. At the tract level, ANCOVAs revealed significant omnibus effects of group on FA in the CCMa (*F*(2,39) = 4.30, *p* = 0.021), CCMi (*F*(2,40) = 4.76, *p* = 0.014), left IFOF (*F*(2,38) = 5.71, *p* = 0.007), right IFOF (*F*(2,37) = 6.87, *p* = 0.003), and left ILF (*F*(2,39) = 5.26, *p* = 0.009). For all of these tracts, the lowest mean FA was observed in the XRT group while the CTL group showed the highest mean FA. Individual group comparisons revealed significant FA differences between XRT and PRT groups (all XRT < PRT) in the left IFOF (*t*(38) = −2.78, *p* = 0.008), right IFOF (*t*(37) = −2.06, *p* = 0.047), left ILF (*t*(39) = −2.62, *p* = 0.013), and right UNC (*t*(39) = −2.06, *p* = 0.047). No significant differences were found between PRT and CTL groups for any tract. A supplemental analysis including only focal radiotherapy patients found similar patterns of tract-level group differences in FA. FA was significantly lower in the XRT than the PRT group for the left CB (*t*(29) = −2.62, *p* = 0.014), left IFOF (*t*(29) = −2.30, *p* = 0.029), and left ILF (*t*(30) = −3.13, *p* = 0.004) while no significant differences were found between PRT and CTL groups (Appendix A).

Similar patterns were identified for RD, with significant omnibus group effects in the CCMa (*F*(2,39) = 3.48, *p* = 0.041), CCMi (*F*(2,40) = 7.06, *p* = 0.002), left IFOF (*F*(2,38) = 8.04, *p* = 0.001), right IFOF (*F*(2,37) = 6.87, *p* = 0.003), left ILF (*F*(2,30) = 11.00, *p* < 0.001), right ILF (*F*(2,37) = 5.18, *p* = 0.010), and right UNC (*F*(2,39) = 4.43, *p* = 0.018). For all but one tract, mean RD was highest for the XRT group and lowest for the CTL group (except R UNC, where CTL > PRT). Individual group comparisons revealed significantly higher RD in XRT than PRT patients for left IFOF (*t*(38) = 2.81, *p* = 0.008), left ILF (*t*(39) = 2.88, *p* = 0.006), right ILF (*t*(37) = 2.37, *p* = 0.023), and right UNC (*t*(39) = 2.40, *p* = 0.021). The only significant PRT-CTL difference in RD was found in the right IFOF, with significantly higher RD in the PRT group (*t*(37) = −2.05, *p* = 0.047).

Regarding AD, a significant omnibus group effect was found only for the right SLF, such that AD was highest in the PRT group and lowest in the CTL group (*F*(2,40) = 4.65, *p* = 0.015). This was driven by a significant difference between PRT and CTL groups (PRT > CTL; *t*(40) = 2.95, *p* = 0.005). No differences in AD for any tract were found between XRT and PRT groups.

### 3.3. Supplemental Analyses: Time since Radiation and White Matter Integrity

Significant group differences in FA and RD were followed up with supplemental within-group analyses to determine whether time since radiotherapy (RT time) may account for differences between XRT and PRT groups. Specifically, if group differences were attributable to RT time, we may expect this variable to predict decreased FA and increased RD within each treatment group; this approach avoids redundancy between group and time since treatment. Linear mixed models with RT time as a fixed effect and subjects as a random effect found that across all tracts, the relationship between RT time and FA was not significant for the XRT group (*t*(109) = 0.85, *p* = 0.398) or the PRT group (*t*(137) = −0.36, *p* = 0.718). Similarly, there was no significant association between RT time and RD in PRT patients (*t*(137) = 0.56, *p* = 0.577). In XRT patients, longer RT time predicted lower RD (*t*(109) = −2.36, *p* = 0.020), which is opposite to what one might expect based on between-group findings (i.e., XRT group showed higher RD and longer time since treatment).

### 3.4. Group Differences in Neuropsychological Performance

When covarying for age and handedness, the XRT group scored significantly lower than the PRT group across all measures of cognitive and motor functioning (Table 3; all *t*(40) < −2.37, all *p* < 0.023). In contrast, no significant differences were observed between PRT and CTL groups for any measure (all *p* > 0.191). Across all neuropsychological measures except PRI, mean standard scores for the XRT group were more than 1 standard deviation below the normative mean of 100 (i.e., < 85), whereas all mean scores for the PRT and CTL groups were above 85. Visual–motor skills and motor coordination (also expressed as standard scores) were the most impaired domains for the XRT group (VMI mean = 69.0, MC mean = 66.8). However, these areas were relative weaknesses for PRT and CTL groups as well, despite mean scores being within normal limits. A supplemental analysis, including only focal radiotherapy patients, revealed significantly lower overall intelligence, verbal reasoning, visual-motor skills, and motor coordination in the XRT group relative to the PRT group. No significant differences were found between PRT and CTL groups (Appendix A).

### 3.5. Relationships between Neuropsychological Performance and White Matter Integrity

Pearson correlation revealed significant associations between neuropsychological test performance and FA of several white matter tracts. Cognitive and motor skills were most consistently associated with the CCMa and left ILF, each of which showed significant positive associations between FA and 4 out of 7 cognitive domains. Specifically, increased FA in the CCMa was associated with higher FSIQ, WMI, PSI, and MC scores (all *r* > 0.30, all *p* < 0.038), while increased FA of the left ILF was associated with higher FSIQ, VCI, PSI, and MC scores (all *r* > 0.37, all *p* < 0.012). Additionally, FA of the left IFOF was positively associated with MC (*r* = 0.39, *p* = 0.010), and FA of the right CB was negatively associated with PSI (*r* = −0.34, *p* = 0.028). Statistically significant findings are summarized in Table 4.

## 4. Discussion

Conventional radiotherapy for pediatric brain tumors has been associated with both white matter compromise and cognitive decline. Recent evidence suggests that cognitive outcomes are more favorable for children treated with PRT compared to those who received XRT [11,49,50,54]. Preservation of white matter has been proposed as a major mechanism of cognitive sparing in PRT, but direct evidence for this is limited. To this end, this exploratory study is the first to directly compare white matter integrity following XRT versus PRT for pediatric brain tumors and to investigate the relationship between cognitive functioning and white matter microstructural integrity in this population.

### 4.1. Reduced White Matter Integrity following XRT, but Not PRT

As anticipated, white matter was broadly compromised following treatment with XRT, whereas microstructural integrity was similar in PRT and CTL groups. Of note, most tracts showed a stepwise pattern across groups, with the greatest white matter compromise in the XRT group, the least in the CTL group, and the PRT group falling in between. In general, differences between PRT and CTL groups were smaller than those observed between the two treatment groups (i.e., XRT vs. PRT). However, for some tracts (e.g., CCMa, CCMi), the stepwise pattern was more evenly distributed, with the clearest differences between the two extremes (i.e., XRT vs. CTL). Sensitivity to small group differences, such as those between the PRT and CTL groups, was limited by sample size; therefore, further investigation is necessary to clarify the extent of white matter damage associated with PRT.

Notably, group differences in white matter structure were reflected in FA and RD but not AD values. FA is considered a sensitive measure of overall integrity, but changes may indicate a wide range of underlying mechanisms. On the other hand, AD and RD measure diffusion along the primary and perpendicular axes of an axon, respectively. Thus, patterns of change across AD and/or RD may imply specific microstructural changes. In this study, reduced FA in the XRT group was consistently driven by an increase in RD without associated changes in AD. The literature suggests that demyelination may underlie this pattern [67,68], which is consistent with the evidence of progressive demyelination following conventional radiotherapy [29], though we acknowledge that multiple, and potentially competing, forms of pathology may be underlying diffusion-related metrics.

Taken together, this study provides preliminary support for a demyelinating mechanism of white matter disruption following radiotherapy as at least one source of neural injury and cognitive impairment. PRT may limit these changes compared to XRT; while early evidence is mixed regarding neuroprotective pharmacological treatments [69], these may eventually offer another avenue of injury prevention. Efforts to promote remyelination are an important area of ongoing research in the treatment of other conditions, such as multiple sclerosis and traumatic brain/spinal cord injury (for review, [70]). These therapies, although still in development, may one day have further significant benefits for pediatric brain tumor survivors.

### 4.2. Tract-Specific Findings

This study found the most consistent group differences in white matter integrity involving the corpus callosum and bilateral IFOF and ILF and the right UNC. Each of these is a major fiber tract responsible for coordinating a complex, distributed network of functional systems. FA in these pathways tends to increase most rapidly during childhood and early adolescence, then level off during late adolescence and early adulthood [71,72]. Moreover, these pathways have been implicated in a wide range of cognitive functions, including many not directly examined in this study, such as executive functioning [73], attention [74], language and semantic processing [75,76,77] memory [78], and social cognition [79]. Future research may specifically target these tracts and associated functions to better determine how these domains relate to regional patterns of white matter change following radiotherapy.

### 4.3. Time since Treatment Does Not Predict Within-Group White Matter Integrity

Group differences may be interpreted as white matter sparing in PRT; however, all XRT patients were farther from treatment than PRT patients. Therefore, reduced white matter integrity in XRT may reflect additional time for progression of late effects. To investigate this, we examined associations between time since treatment and white matter integrity within each group. Within both groups, longer time since treatment did not predict more white matter damage. In fact, in the XRT group, time since treatment was associated with *lower* RD (i.e., improved integrity). This was unexpected and should be interpreted with caution, but could potentially indicate some degree of long-term remyelination [80]. Because myelination continues throughout adulthood, those treated with radiotherapy as children may have some capacity to repair white matter damage despite their altered trajectory of neurodevelopment.

### 4.4. Cognitive Sparing in PRT

In line with previous research, the current study found significantly lower neuropsychological test scores in those treated with XRT compared to the PRT group [49,50,53,54]. This was the case across all domains, including overall intellectual functioning, verbal and perceptual reasoning, working memory, processing speed, visual-motor integration, and motor coordination. Paralleling neuroimaging findings, no significant differences were found between the PRT and CTL groups on any cognitive measures. In the XRT group, all domains except perceptual reasoning fell at least one standard deviation below the normative mean, with the greatest impairment found in visual–motor skills and motor coordination. On the other hand, the PRT and CTL groups consistently performed within one standard deviation of typical expectations. These findings add to the growing literature demonstrating improved neurocognitive outcomes in PRT compared to XRT. Of note, although processing speed and motor coordination were within normal limits for the PRT group (mean scores of 89.2 and 85.3, respectively), these were considered to be in the “low average” range, and were relatively weaker than other domains. Therefore, although outcomes are significantly improved relative to those treated with XRT, these specific areas remain vulnerable and may be appropriate targets for rehabilitation.

As described above, intellectual and psychomotor functioning have been consistently associated with white matter integrity following radiation [10,39,44,45,81]. In the current study, these domains were all positively associated with FA, specifically of the CCMa and left ILF. Across participants in all groups, these tracts most consistently predicted neuropsychological performance. These exploratory findings offer preliminary support for a direct association between white matter damage and neurocognitive outcomes, as well as shed light on the roles of specific tracts that may be further explored in future studies.

### 4.5. Limitations and Future Directions

This was a small, exploratory study of pediatric brain tumor survivors. As with all studies of individuals with brain tumors, our study sample reflected considerable heterogeneity in type, location, and size of tumors. Moreover, the primary effects of disease cannot be fully disentangled from treatment effects. However, in the current study, similarities between PRT and Control groups strongly implicate treatment differences in cognitive and white matter sparing.

Another major limitation of this study is the non-random assignment of patients to treatment groups. As described above, all patients treated before 2007 received XRT, whereas those treated since 2007 received PRT as the standard of care. Recruitment for this study did not begin until 2018; therefore, the XRT group was limited in size, older, and farther out from treatment than the PRT group. Supplemental within-group analyses did not suggest a direct relationship between time since treatment and DTI findings; however, cohort effects cannot be entirely ruled out and require further investigation. Our group is currently working to compare cognitive outcomes in patients concurrently treated with XRT versus PRT according to the standard of care in an international, multisite study.

In addition to radiation type, treatment variables, such as craniospinal irradiation (CSI), chemotherapy, and hydrocephalus, confer additional cognitive risk. Supplemental analyses excluding CSI patients yielded similar results to primary analyses. However, these findings must be interpreted with caution due to the very small sample of focal-only patients (i.e., 5 XRT, 8 PRT). Future studies should also include dosimetry associations. Detailed examination of all treatment-related variables is beyond the scope of this exploratory study, but it will be important for future research with larger samples to examine these confounding variables. Specifically, chemotherapy agents commonly prescribed as part of brain tumor treatment protocols (e.g., intrathecal methotrexate) have been associated with white matter change and cognitive deficits, especially in very young children [82].

The exploratory nature of this study resulted in group comparisons across a large number of white matter tracts. Multiple comparison corrections were not conducted given the small sample size; therefore, findings for specific tracts should be interpreted with caution. However, overall group comparisons were conducted through linear mixed modeling, which accounts for non-independence across tracts in a single model. Therefore, broad patterns across all tracts may be interpreted with more confidence. Finally, the location of the radiation field is likely associated with the spatial distribution of white matter compromise; this is beyond the scope of the current study, but it is an important question for further examination. The preliminary findings reported in this study may serve as guidance for future research, which may include larger, more diagnostically homogenous samples and more focused regions of interest.

## 5. Conclusions

This study offers preliminary evidence that white matter preservation may underlie long-term cognitive sparing following PRT compared to XRT for pediatric brain tumor patients. Both white matter structural integrity and neurocognitive scores were significantly lower in those treated with XRT compared to PRT while patients treated with PRT showed similar white matter integrity and neuropsychological test performance to healthy controls. Moreover, the integrity of the CCMa and left ILF emerged as the most consistent predictors of general intellectual ability and psychomotor speed/control. This study adds to the growing literature demonstrating long-term advantages of PRT for treating pediatric brain tumors. Moreover, the development of effective neuroprotective and remyelinating treatments may further mitigate the cognitive and functional impacts of radiotherapy in the future.

## Figures and Tables

**Figure 1 cancers-15-01844-f001:**
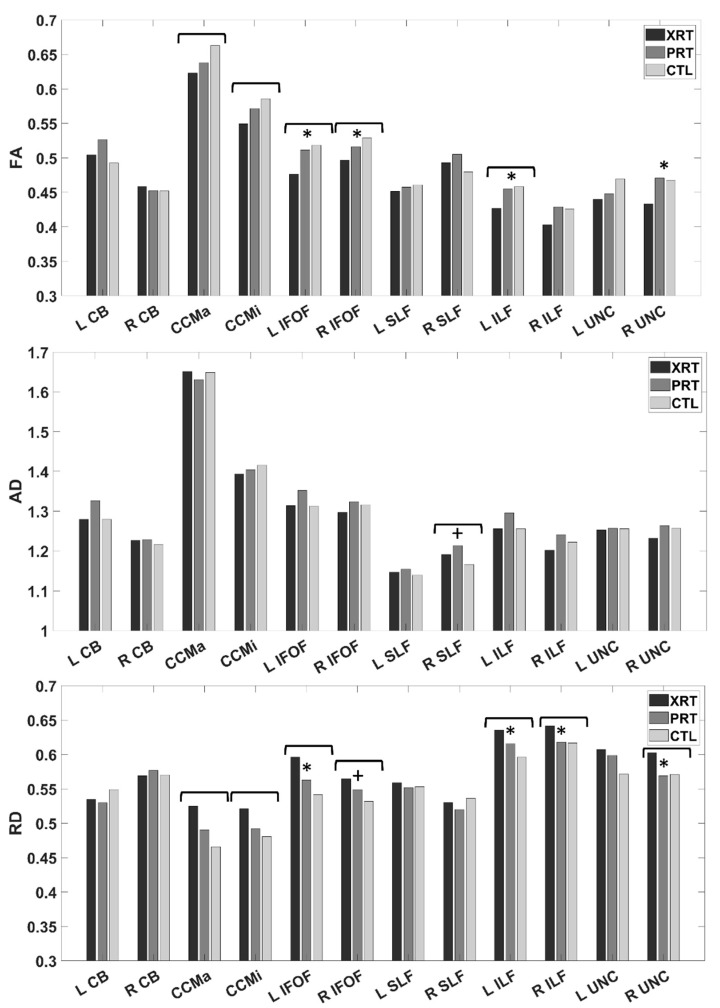
Unadjusted group means are shown for each tract for FA (top), AD (middle), and RD (bottom). Group differences were determined by ANCOVA with age and handedness as covariates, with α = 0.05. Significant omnibus group differences (XRT-PRT-CTL) are depicted by brackets. Significant XRT-PRT differences are depicted by *. Significant PRT-CTL differences are depicted by +.

**Table 1 cancers-15-01844-t001:** Demographic and Clinical Characteristics of Participants.

	XRT (*n* = 10)		PRT (*n* = 12)		CTL (*n* = 23)		Χ^2^	*p*
	*n*	%	*n*	%	*n*	%		
**Sex**							1.97	0.373
Male	7	70	6	50	10	43		
Female	3	30	6	50	13	57		
**Handedness**							8.84	0.012
Right	10	100	9	75	23	100		
Left	0	0	3	25	0	0		
**Race**							0.41	0.814
White	9	90	10	83	19	83		
Black	1	10	2	17	2	9		
Unknown	0	0	0	0	2	9		
**Ethnicity**							1.47	0.479
Hispanic/Latino	5	50	3	25	8	35		
Not Hispanic/Latino	5	50	9	75	13	57		
Unknown	0	0	0	0	2	9		
**Maternal Education**							4.39	0.625
<High school	0	0	2	17	1	4		
High school	6	60	5	42	12	52		
4-year college degree	1	10	4	33	4	17		
Advanced degree	1	10	1	8	3	13		
Unknown	2	20	0	0	3	13		
**Family Income ($)**							10.26	0.114
<40,000	2	20	2	17	8	35		
40,000–79,999	3	30	3	25	8	35		
80,000+	3	30	7	58	7	30		
Unknown	2	20	0	0	0	0		
**Tumor location**							0.22	0.639
Supratentorial	4	40	6	50	---			
Infratentorial	6	60	6	50	---			
**Tumor type**							0.35	0.950
Low Grade Glioma	2	20	3	25	---			
Embryonal Tumor	4	40	4	33	---			
Ependymoma	1	10	2	17	---			
Other	3	30	3	25	---			
**RT technique**								
CSI	5	50	4	33	---		0.63	0.429
Focal	5	50	8	67	---			
**Ventriculoperitoneal Shunt**							1.56	0.211
Yes	6	60	4	33	---			
No	4	40	8	67	---			
**Chemotherapy**							3.32	0.069
Yes	8	80	5	42	---			
No	2	20	7	58	---			
	**XRT (*n* = 10)**		**PRT (*n* = 12)**		**CTL (*n* = 23)**		** *F* **	** *p* **
	Mean (SD)	Min-Max	Mean (SD)	Min-Max	Mean (SD)	Min-Max		
**Age at evaluation (yrs)**	21.7 (5.7)	15.3–34.5	16.9 (4.6)	10.4–23.7	15.5 (5.3)	6.8–29.3	4.91	0.012
**Household size**	4.4 (1.2)	2–6	4.3 (1.2)	3–7	4.8 (1.5)	2– 8	0.67	0.517
**Age at diagnosis (yrs)**	5.9 (3.8)	0.8–12.7	7.1 (4.2)	1.8–16.1	---		0.52	0.479
**Time since RT (yrs)**	14.7 (2.4)	12.2–18.5	8.9 (1.5)	7.1–11.8	---		47.65	<0.001
**Total RT dose for primary tumor (cGy)**	5338 (370)	4500–5940	5355 (308)	5040–5940	---		0.01	0.908
**# of Craniotomies**	1.3 (0.48)	1–2	1.2 (1.1)	0–4	---		0.12	0.730
**Karnofsky–Lansky**	72.9 (13.8)	50.0–90.0	84.5 (16.9)	50.0–100.0	---		2.33	0.147

Missing data include Household Size (*n =* 2 PRT) and Karnofsky–Lansky score (*n* = 3 XRT, 1 PRT). Karnofsky–Lansky scores are from first post-operative appointment.

**Table 2 cancers-15-01844-t002:** Results of linear mixed models.

	XRT vs. PRT			PRT vs. CTL		
	*β*	*t*	*p*	*β*	*t*	*p*
FA	−0.027	−2.58	**0.010**	0.006	0.65	0.515
AD	−0.004	−0.19	0.853	−0.019	−1.20	0.229
RD	0.035	4.21	**<0.001**	−0.012	−1.80	0.073

*β*, *t*, and *p*-values are derived from linear mixed models with handedness and age as fixed covariates and subject as a random variable. Bold text indicates *p* < 0.05.

**Table 3 cancers-15-01844-t003:** Neuropsychological Test Scores.

	XRT (*n* = 10)		PRT (*n* = 12)		CTL (*n* = 23)	
	Mean (SD)	Min-Max	Mean (SD)	Min-Max	Mean (SD)	Min-Max
FSIQ	80.0 (14.5)	61–106	98.6 (14.6)	76–129	99.7 (10.9)	84–125
VCI	83.9 (11.4)	68–107	101.3 (17.1)	76–136	98.1 (12.8)	78–118
PRI	89.6 (16.6)	69–119	103.8 (12.2)	83–125	100.0 (12.3)	79–125
WMI	81.5 (16.2)	58–105	101.0 (15.9)	79–135	102.2 (10.3)	88–122
PSI	75.2 (16.8)	59–103	89.2 (18.1)	68–126	101.0 (11.2)	75–123
VMI	69.0 (20.7	45–103	92.0 (9.4)	78–112	86.5 (11.5)	50–104
MC	66.8 (13.4)	45–89	85.3 (12.6)	64–102	88.1 (11.4)	61–102
	**XRT vs. PRT**		**PRT vs. CTL**			
	*t*	*p*	*t*	*p*		
FSIQ	−3.15	**0.003**	0.59	0.561		
VCI	−2.48	**0.018**	0.15	0.883		
PRI	−2.38	**0.022**	−0.10	0.920		
WMI	−2.84	**0.007**	0.50	0.617		
PSI	−2.52	**0.016**	1.33	0.192		
VMI	−3.10	**0.004**	−0.92	0.363		
MC	−3.83	**<0.001**	−0.06	0.951		

Unadjusted means are presented. *t-* and *p*-values are derived from ANCOVAs with handedness and age as covariates. Bold text indicates *p* < 0.05.

**Table 4 cancers-15-01844-t004:** Significant Correlations Between Tract-Level FA and Neuropsychological Test Scores.

	R CB		CCMa		L IFOF		L ILF	
# of Domains	1/7		4/7		1/7		4/7	
	*r*	*p*	*r*	*p*	*r*	*p*	*r*	*p*
FSIQ	**---**	**---**	0.39	0.010	**---**	**---**	0.38	0.011
VCI	**---**	**---**	**---**	**---**	**---**	**---**	0.41	0.006
PRI	**---**	**---**	**---**	**---**	**---**	**---**	**---**	**---**
WMI	**---**	**---**	0.31	0.037	**---**	**---**	**---**	**---**
PSI	−0.34	0.028	0.51	<0.001	**---**	**---**	0.44	0.003
VMI	**---**	**---**	**---**	**---**	**---**	**---**	**---**	**---**
MC	**---**	**---**	0.41	0.005	0.39	0.010	0.39	0.009

## Data Availability

Data are available upon request.

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
