# Peer review of "Cognitive Sparing in Proton versus Photon Radiotherapy for Pediatric Brain Tumor Is Associated with White Matter Integrity: An Exploratory Study"

_cancers, 2023, doi:10.3390/cancers15061844_

Round 1

Reviewer 1 Report

Summary: This is a study of >20 patients who received radiation therapy as children for brain tumors.  The authors compared patients who received photon (XRT) vs. proton (PRT) therapy.  Of note PRT was done in a later era.

Impact: This is an interesting exploratory study.  The results examine important late side effects of radiation therapy.  There are likely major confounders, including time since treatment (which the authors examined), and whether CSI was done (which the authors need to examine more carefully).  The results are interpreted with caution, appropriately.  Images to provide examples of the DTI imaging and tract findings, and the tumor locations and/or radiation fields, would be helpful.  A number of concerns are further detailed below:

Major concerns:

1.     CSI is likely to have a major impact on neurocognitive function, which may be somewhat independent of proton vs. photon technique.  And CSI is more frequent in the XRT group, which is a major confounder.  In fact both PRT and XRT would be expected to deliver considerable dose to normal brain with CSI.  Can the authors perform a subgroup analysis of the non-CSI patients?  Is there still a signal for a difference between XRT (non-CSI) and PRT (non-CSI)?  This would help determine if CSI was confounding the results.

Minor concerns:

1.     The grammar seems off in several places.  For instance “history of brain tumor” on line 28-29 and also 31 and in several other areas should be “history of brain tumors” (plural).

2.     Again the grammar need to be checked throughout.  For instance Line 126: “part of a several ongoing studies…” should remove the “a” here.

3.     Line 126: please specify the ongoing studies.  Which studies?  How many total studies?  How were the patients selected?

4.     Can some kind of image of the FA, RD, AD tractography for each group be shown, showing representative differences?

5.     Can the authors provide images of the location of the different tumors for all patients?  This would give readers insight into where the radiation fields may have been targeted.

6.  Can any of the anatomic areas of the imaging findings be correlated with the specific location of the radiation field?  For instance were radiation fields in the right frontal lobe associated with changes in tracts in the right frontal lobe?

Author Response

Reviewer 1:

Comments and Suggestions for Authors

Summary: This is a study of >20 patients who received radiation therapy as children for brain tumors.  The authors compared patients who received photon (XRT) vs. proton (PRT) therapy.  Of note PRT was done in a later era.

Impact: This is an interesting exploratory study.  The results examine important late side effects of radiation therapy.  There are likely major confounders, including time since treatment (which the authors examined), and whether CSI was done (which the authors need to examine more carefully).  The results are interpreted with caution, appropriately.  Images to provide examples of the DTI imaging and tract findings, and the tumor locations and/or radiation fields, would be helpful.  A number of concerns are further detailed below:

Major concerns:

  1. CSI is likely to have a major impact on neurocognitive function, which may be somewhat independent of proton vs. photon technique.  And CSI is more frequent in the XRT group, which is a major confounder.  In fact both PRT and XRT would be expected to deliver considerable dose to normal brain with CSI.  Can the authors perform a subgroup analysis of the non-CSI patients?  Is there still a signal for a difference between XRT (non-CSI) and PRT (non-CSI)?  This would help determine if CSI was confounding the results.

Response: We agree that CSI is an important risk factor for neurocognitive outcomes following radiotherapy that warrants additional consideration in this study. As suggested, we conducted additional analyses excluding patients in both treatment groups who underwent CSI; the results of these analyses are now provided as supplemental materials (Figure S2, Tables S2-S3). These findings must be interpreted with caution, due to the very small remaining sample size of each treatment group (XRT n = 5, PRT n = 8). Nonetheless, the pattern of group differences did not substantially change for either neuroimaging findings or neuropsychological test scores when excluding CSI patients.

Specifically, the results of linear mixed models again revealed significant differences in FA (XRT < PRT) and RD (XRT > PRT) between treatment groups, but not between PRT and control groups (Table S2). Similar patterns were also found for tract-level FA analyses, despite the small sample size (Figure S2). The PRT group continued to outperform the XRT group across all neuropsychological domains, although only 4/7 comparisons reached statistical significance (i.e., FSIQ, VCI, VMI, Motor Coordination). Again, no statistically significant differences were found between the PRT and control groups with respect to cognitive functioning (Table S3).

The following text has been added to the manuscript:

Page 7 (Line 267):

“To examine potential effects of craniospinal irradiation on these findings, supplemental analyses were performed including only patients who underwent focal radiotherapy (XRT n = 5, PRT n = 8). Linear mixed models in this subsample of participants yielded the same pattern of results, with the XRT group showing significantly lower overall FA and higher overall RD compared to the PRT group. Again, no significant differences between PRT and CTL groups were found (Supplemental Table S2).

Page 8 (Line 286):

“Supplemental analysis including only focal radiotherapy patients found similar patterns of tract-level group differences in FA. FA was significantly lower in the XRT than the PRT group for the left CB (t(29) = -2.62, p = .014), left IFOF (t(29) = -2.30, p = .029), and left ILF (t(30)= -3.13, p = .004), while no significant differences were found between PRT and CTL groups (Supplemental Figure S2).”

Page 10 (Line 334):

“A supplemental analysis including only focal radiotherapy patients revealed significantly lower overall intelligence, verbal reasoning, visual-motor skills, and motor coordination in the XRT group relative to the PRT group. No significant differences were found between PRT and CTL groups (Supplemental Table S3).”

Page 13, Lines 499- also in response to Reviewer 3’s feedback:

“In addition to radiation type, treatment variables such as craniospinal irradiation (CSI), systemic chemotherapy, and hydrocephalus confer additional cognitive risk. Supplemental analyses excluding CSI patients yielded similar results to primary analyses. However, these findings must be interpreted with caution due to the very small sample of focal-only patients (i.e., 5 XRT, 8 PRT). Future studies should also include dosimetry associations. Detailed examination of all treatment-related variables is beyond the scope of this exploratory study, but it will be important for future research with larger samples to examine these confounding variables. Specifically, chemotherapy agents commonly prescribed as part of brain tumor treatment protocols (e.g., intrathecal methotrexate) have been associated with white matter change and cognitive deficits, especially in very young children [82]”

Minor concerns:

  1. The grammar seems off in several places.  For instance “history of brain tumor” on line 28-29 and also 31 and in several other areas should be “history of brain tumors” (plural).
  2. Again the grammar need to be checked throughout.  For instance Line 126: “part of a several ongoing studies…” should remove the “a” here.

Response: We thank the reviewer for their careful attention to detail. These grammatical errors have been corrected.

  1. Line 126: please specify the ongoing studies.  Which studies?  How many total studies?  How were the patients selected?

Response: This description of enrollment was an error, as all of these participants were recruited from the same study protocol. The sentence has been revised to read: “Participants were enrolled on this study examining neuroimaging and cognitive outcomes in long-term survivors of pediatric brain tumor.” Selection and exclusion criteria are further detailed in the following two paragraphs (Section 2.1)

  1. Can some kind of image of the FA, RD, AD tractography for each group be shown, showing representative differences?

Response: Unfortunately, the tractography approach used in this study (AFQ, Yeatman et al., 2012) estimates fiber tracts in each subject’s native space, resulting in unique tracts for each individual. Therefore, tracts cannot be averaged together into a single image for each group.

We also considered showing images from a single subject for each group (PRT, XRT, CTL); however, high variability within groups made it difficult to identify a “representative” subject and to appreciate group differences at the individual level.  However, we agree that a visual representation of the tracts included in this study would be helpful to orient readers. To this end, we have added a figure (Figure S1) showing all tracts of interest in a control participant.

Updates to the text: Line 214:

“The AFQ method tracks fibers in each individual’s native space, resulting in subject-specific tracts. All tracts are shown in a representative control participant in Figure S1.”

  1. Can the authors provide images of the location of the different tumors for all patients?  This would give readers insight into where the radiation fields may have been targeted.

Response: We thank the author for this suggestion and agree that tumor location data would be informative. To make this information clearest, we have provided a supplemental table (Table S1) including each patient’s tumor location based on their first MRI prior to resection.

  1. Can any of the anatomic areas of the imaging findings be correlated with the specific location of the radiation field?  For instance were radiation fields in the right frontal lobe associated with changes in tracts in the right frontal lobe?

Response: We agree with this reviewer that this would be highly useful in interpreting our findings, and this is a high priority for future investigation. Unfortunately, for this dataset we do not have access to masks of specific radiation fields for each participant to compare with the location of each tract. This has been added as another potential direction for future research in the Discussion section (Line 477):

“Finally, the location of the radiation field is likely associated with the spatial distribution of white matter compromise; this is beyond the scope of the current study, but is an important question for further examination.”

Reviewer 2 Report

In this Manuscript, the authors examined the relationship between white matter change and late cognitive effects in pediatric brain tumor survivors treated with a photon (XRT) versus proton radiotherapy (PRT). 

The authors have found that white matter integrity and neuropsychological performance were lower in XRT patients, while in PRT patients were similar to control participants. 

The manuscript is well-structured, and clearly written, this reviewer enjoyed reading it.  Experimental design is appropriate to test the hypothesis, the authors have answered to all my questions sooner or later in the Manuscript.

The tables are easy to understand and are well designed. 

However, a problematic spacing between some words was spotted and should be corrected. 

Author Response

Reviewer 2:

Comments and Suggestions for Authors

In this Manuscript, the authors examined the relationship between white matter change and late cognitive effects in pediatric brain tumor survivors treated with a photon (XRT) versus proton radiotherapy (PRT). 

The authors have found that white matter integrity and neuropsychological performance were lower in XRT patients, while in PRT patients were similar to control participants. 

The manuscript is well-structured, and clearly written, this reviewer enjoyed reading it.  Experimental design is appropriate to test the hypothesis, the authors have answered to all my questions sooner or later in the Manuscript.

The tables are easy to understand and are well designed. 

However, a problematic spacing between some words was spotted and should be corrected.

Response: We thank this reviewer for their positive comments. We have corrected the spacing between words to be uniform across the manuscript.

Reviewer 3 Report

The exploratory study was designed to investigate the relationship between white matter damage and late cognitive effects in pediatric brain tumor survivors who were treated with photon or proton therapy.

1.     Introduction is too long and contains repetitions

2.     Provide a proper quotation for the statement “Research suggests that PRT is similarly effective but offers improved long-term cognitive outcomes compared to XRT” -line 97

3.     As beyond the radiotherapy modality (PRT versus XRT, the radiotherapy dose and the volume), several other risk factors may lead to worse chronic neurocognitive effects (e.g longer time since diagnosis, systemic chemotherapy, as well as clinical variables such as hydrocephalus), relevant differences between the treatment groups should be mentioned. Regarding chemotherapy, impact of it on neurodevelopment should be considered and perhaps discussed.

Author Response

Reviewer 3:

Comments and Suggestions for Authors

The exploratory study was designed to investigate the relationship between white matter damage and late cognitive effects in pediatric brain tumor survivors who were treated with photon or proton therapy.

  1. Introduction is too long and contains repetitions

Response: The introduction has been revised to eliminate a detailed description of diffusion tensor imaging, which is provided later in the methods section, and to remove repetitive information about specific cognitive domains impacted by radiotherapy and white matter decline.

  1. Provide a proper quotation for the statement “Research suggests that PRT is similarly effective but offers improved long-term cognitive outcomes compared to XRT” -line 97

Response: The appropriate citations have been added ([48-50]).

  1. As beyond the radiotherapy modality (PRT versus XRT, the radiotherapy dose and the volume), several other risk factors may lead to worse chronic neurocognitive effects (e.g longer time since diagnosis, systemic chemotherapy, as well as clinical variables such as hydrocephalus), relevant differences between the treatment groups should be mentioned. Regarding chemotherapy, impact of it on neurodevelopment should be considered and perhaps discussed.

Response: This reviewer raises excellent points about important clinical factors to consider in this population. Time since diagnosis is the most considerable limitation in our sample, given that XRT and PRT participants underwent treatment at different times (i.e., before or after 2007). As Reviewer 1 pointed out, this group difference is discussed in the manuscript and examined through supplemental, within-group analyses (described on page 9-10, Section 3.3). Our groups did not significantly differ with respect to chemotherapy or VP shunting, but these are also important clinical variables that may impact cognitive outcomes. This is now discussed in section 4.5 (Limitations and Future Directions), line 499:

“In addition to radiation type, treatment variables such as craniospinal irradiation (CSI), systemic chemotherapy, and hydrocephalus confer additional cognitive risk. Supplemental analyses excluding CSI patients yielded similar results to primary analyses. However, these findings must be interpreted with caution due to the very small sample of focal-only patients (i.e., 5 XRT, 8 PRT). Future studies should also include dosimetry associations. Detailed examination of all treatment-related variables is beyond the scope of this exploratory study, but it will be important for future research with larger samples to examine these confounding variables. Specifically, chemotherapy agents commonly prescribed as part of brain tumor treatment protocols (e.g., intrathecal methotrexate) have been associated with white matter change and cognitive deficits, especially in very young children [82]”

Round 2

Reviewer 1 Report

The authors have thoughtfully addressed my concerns.  I think the title should read "pediatric brain tumors" instead of "pediatric brain tumor".